# Large-scale analysis of whole genome sequencing data from formalin-fixed paraffin-embedded cancer specimens demonstrates preservation of clinical utility

Shadi Basyuni[1,2,11], Laura Heskin[1,2,11], Andrea Degasperi[1,2], Daniella Black[1,2], Gene C. C. Koh[1,2], Lucia Chmelova[1,2], Giuseppe Rinaldi[1,2], Steven Bell[3,4], Louise Grybowicz[5], Greg Elgar[6], Yasin Memari[1,2], Pauline Robbe[7,8], Zoya Kingsbury[9], Carlos Caldas[4], Jean Abraham[3,4], Anna Schuh[7], Louise Jones[10], PARTNER Trial Group*, Personalised Breast Cancer Program Group*, Marc Tischkowitz[1], Matthew A. Brown[7], Helen R. Davies[1,2] & Serena Nik-Zainal[1,2] ✉

Whole genome sequencing (WGS) provides comprehensive, individualised cancer genomic information. However, routine tumour biopsies are formalin-fixed and paraffin-embedded (FFPE), damaging DNA, historically limiting their use in WGS. Here we analyse FFPE cancer WGS datasets from England's 100,000 Genomes Project, comparing 578 FFPE samples with 11,014 fresh frozen (FF) samples across multiple tumour types. We use an approach that characterises rather than discards artefacts. We identify three artefactual signatures, including one known (SBS57) and two previously uncharacterised (SBS FFPE, ID FFPE), and develop an "FFPEImpact" score that quantifies sample artefacts. Despite inferior sequencing quality, FFPE-derived data identifies clinically-actionable variants, mutational signatures and permits algorithmic stratification. Matched FF/FFPE validation cohorts shows good concordance while acknowledging SBS, ID and copy-number artefacts. While FF-derived WGS data remains the gold standard, FFPE-samples can be used for WGS if required, using analytical advancements developed here, potentially democratising whole cancer genomics to many.

Genome sequencing technologies have advanced our understanding of cancer, providing more personalised management options[1]. Until recently, targeted sequencing strategies, limited to several hundred known cancer-associated genes, have been the mainstay of oncology[2–4].

However, whole genome sequencing (WGS) carries several thousand-fold more data per patient, providing a holistic picture of the cancer genomic state, revealing a multitude of highly individualised, clinically valuable genomic characteristics missed by targeted approaches[5,6].

[1]Department of Medical Genetics, National Institute for Health Research Cambridge Biomedical Research Centre, University of Cambridge, Cambridge, UK. [2]Early Cancer Institute, Department of Oncology, University of Cambridge, Cambridge, UK. [3]Precision Breast Cancer Institute, Department of Oncology, University of Cambridge, Cambridge, UK. [4]Cancer Research UK Cambridge Centre, University of Cambridge, Li Ka Shing Centre, Cambridge, UK. [5]Cambridge Cancer Trials Centre, Department of Oncology, Cambridge University Hospitals, Cambridge, UK. [6]Genomics England, One Canada Square, London, UK. [7]Department of Oncology, University of Oxford, Oxford, UK. [8]RIKEN Centre for Integrative Medical Sciences, Yokohama, Japan. [9]Illumina Cambridge, Ltd, Illumina Centre, Cambridge, UK. [10]Barts Cancer Institute, Queen Mary University of London, Charterhouse Square, London, UK. [11]These authors contributed equally: Shadi Basyuni, Laura Heskin. *Lists of authors and their affiliations appear at the end of the paper. ✉e-mail: sn206@cam.ac.uk

Recent advances in sequencing methodologies have significantly reduced the base cost of sequencing, contributing to the potential for more widespread adoption of WGS in research and clinical settings[7–10]. Yet, a critical barrier preventing utility in the clinical setting is that tumour samples are routinely formalin-fixed and paraffin-embedded (FFPE) for histopathological analysis. The FFPE process causes considerable DNA damage, precluding WGS, which hitherto has been preferentially conducted on fresh frozen (FF) specimens where DNA integrity is preserved. Additionally, the logistical demands associated with snap freezing, including requiring liquid nitrogen equipment and specialist sample handling, prohibit tumour collections through routine diagnostic pathways, particularly in non-tertiary hospital settings and lower-resourced healthcare systems. Targeted genomic approaches are possible on FFPE specimens. Thus they have become the preferred cancer genomic option over the last decade.

Furthermore, previous evaluations of WGS data from FFPE specimens have revealed that most FFPE artefacts are detected at frequencies exceeding background noise (~1% of sequencing reads) but <10%[11–14]. This led to bioinformatic solutions, including filtering variants with allelic fractions of 10% or less. However, this approach leads to the exclusion of genuine mutations, including clinically actionable variants present at low variant allelic fractions (VAFs). Consequently, the value of FFPE WGS has been called into question[15]. Moreover, reports of recurrent false positives have further hindered clinical adoption—for example, a potentially actionable mutation, *EGFR* T790M, has been ascribed as an FFPE-mediated artefact[16]. Mutational signatures have been proposed as a means to identify and quantify artefact[17]; however, they have yet to be comprehensively examined in a substantially powered dataset or validated in clinical cohorts. WGS of FFPE-derived material was thus considered off-limits for many years, with FF or RNAlater-stored[18] samples being the gold standard for WGS.

Here, we set out to explore whether it was possible to salvage WGS data by analysing a large cohort of FFPE-derived specimens. This is driven by the need to extend comprehensive genomics beyond the confines of specialised, well-resourced centres. In addition, there are clinical scenarios where it is simply not possible to obtain FF material, or where the decision for WGS is an ex post facto justification (i.e., one that is made retroactively following, for example, histopathological diagnosis or failure of conventional treatment) and further specimen collections are not possible. Lastly, enabling comprehensive, genome-wide analytics on FFPE-derived tumour material would considerably facilitate exploratory analyses on less well-studied populations, potentially democratising cancer genomics to more diverse cohorts worldwide[19].

## Results

We examined a cohort comprising 578 WGS FFPE-stored treatment naïve primary cancer samples from nine tumour-types and matched germline samples derived from peripheral blood mononuclear cells. These were compared to 11,014 WGS FF samples in the discovery phase. All patients were diagnosed with cancer in England's National Health Service and underwent a primary surgical procedure with curative intent without prior adjuvant therapy. The following tumour-types were available for the study: breast, central nervous system (CNS), colorectal, kidney, lung, ovary, prostate, uterus, and bladder (Supplementary Table 1). A fraction of FF tumour samples underwent polymerase chain reaction (PCR) during library preparation (n = 899, 8% of FF samples, herewith referred to as FF(PCR)). All samples were taken through the standard Genomics England (GEL) WGS bioinformatic pipeline, including alignment and somatic-variant calling using default parameters (Supplementary Methods Section S1).

### Quality of sequencing data

Quality of sequencing data was assessed using customary coverage and alignment metrics (Fig. 1A; Supplementary Table 5 and

Supplementary Data 1). Compared to FF-derived samples, FFPE-derived samples yielded data of poorer quality, with smaller insert sizes (391 base pairs vs. 477 base pairs; $p < 0.0001$) and a higher percentage of chimeric DNA fragments (0.51% vs. 0.26%; $p < 0.0001$), indicative of damaged DNA templates. Mapping of FFPE-derived short reads to the reference genome was marginally reduced (93.4% vs. 94.1%; $p < 0.0001$). There was increased heterogeneity in genome coverage with a bias towards GC over-representation and consequential depletion of AT sequences. These differences were not due to factors such as cancer cell content or tumour type (Supplementary Fig. 1). FF(PCR) metrics interestingly resembled PCR-free FF libraries, implying that the primary cause for poorer sequencing quality in FFPE-derived DNA libraries is the quality of the starting material, and not due to the library-preparation process.

Distributions of VAFs for substitutions and indels were normalised to cancer cell content (nVAF) (Fig. 1B). FF substitution and indel variants demonstrated a peak at nVAF of 0.5, in keeping with the expected heterozygosity of most variants. By contrast, variants from FF(PCR) and FFPE samples showed the tallest peaks at nVAF of 0.1, with a modest peak at nVAF of 0.5 in keeping with PCR-related and/or DNA-damage-induced artefacts. Substitution and structural variation burdens were not systematically increased in FFPE libraries across all tissue types, unlike indel burdens which were increased by order of magnitude irrespective of tumour-type (Fig. 1C). The substantial excess of indels was similarly observed in FF(PCR) libraries, suggesting that the source of indel artefacts could be PCR-related during library preparation.

### Drivers and actionable variants

Domain 1 somatic variants are defined by Genomics England as variants in a virtual panel of potentially actionable genes (168 genes listed in Actionable genes in solid tumour v2; Supplementary Methods Section S1.5). Detection of Domain 1 variants was uncompromised in FFPE-derived samples. There were no significant differences in the prevalence of Domain 1 events (e.g., breast ($p = 0.59$), colorectal ($p = 0.37$), lung ($p = 0.96$), uterus ($p = 0.17$)) between FFPE, FF, and FF(PCR) cohorts (Fig. 2A; Supplementary Fig. 3). As actionable mutations are of immediate clinical value, specific variants were explored. For example, variants associated with gefitinib response in lung cancers were not differentially represented in FFPE-derived WGS samples. *EGFR* variants indicating gefitinib sensitivity (L858R, G719S, or exon 19 deletions) were present in 8.1% (104) of FF lung samples, 8.6% (9) of FF (PCR) samples and 14.1% (9) of FFPE samples. The marginally higher proportion of these EGFR variants in FFPE samples might be attributed to variations in histological subtypes of lung cancer. However, this observation should be interpreted with caution due to the limited sample size. Similarly, the *KRAS* G12C mutation associated with gefitinib resistance was identified in 10.5% (135) of FF samples, 11.5% (12) of FF (PCR) samples, and 7.8% (5) of FFPE samples. Assessing allelic frequency and relative cancer cell content for each of these variants showed no inter-group biases (Fig. 2B, Supplementary Table 6), allaying concerns regarding false positives, as unlike true mutations, artefact would not be expected to correlate with cancer cell content. These findings were consistently observed for actionable variants in other tissue types (e.g., *PIK3CA* in breast cancer, *BRAF* V600E mutation in all organs) (Fig. 2B). Critically, the analysis highlights how previous attempts at bioinformatic filtering of VAF < 10% in FFPE samples would result in the exclusion of true clinical actionable variants. Indeed, 125 (7.7%) of *PIK3CA* and *BRAF* V600E mutations occurred at a VAF < 10% and would have been discarded using VAF filtering. The *EGFR* T790M variant previously purported as an FFPE-mediated artefact was not identified in any of the 578 samples examined, adding further reassurance. Copy number driver events were compared between FF and FFPE groups. Clinically relevant amplification drivers such as *ERBB2* in breast cancers (FF: 7.7%, FFPE:4.7%), *GNAS* (20q13) in colorectal

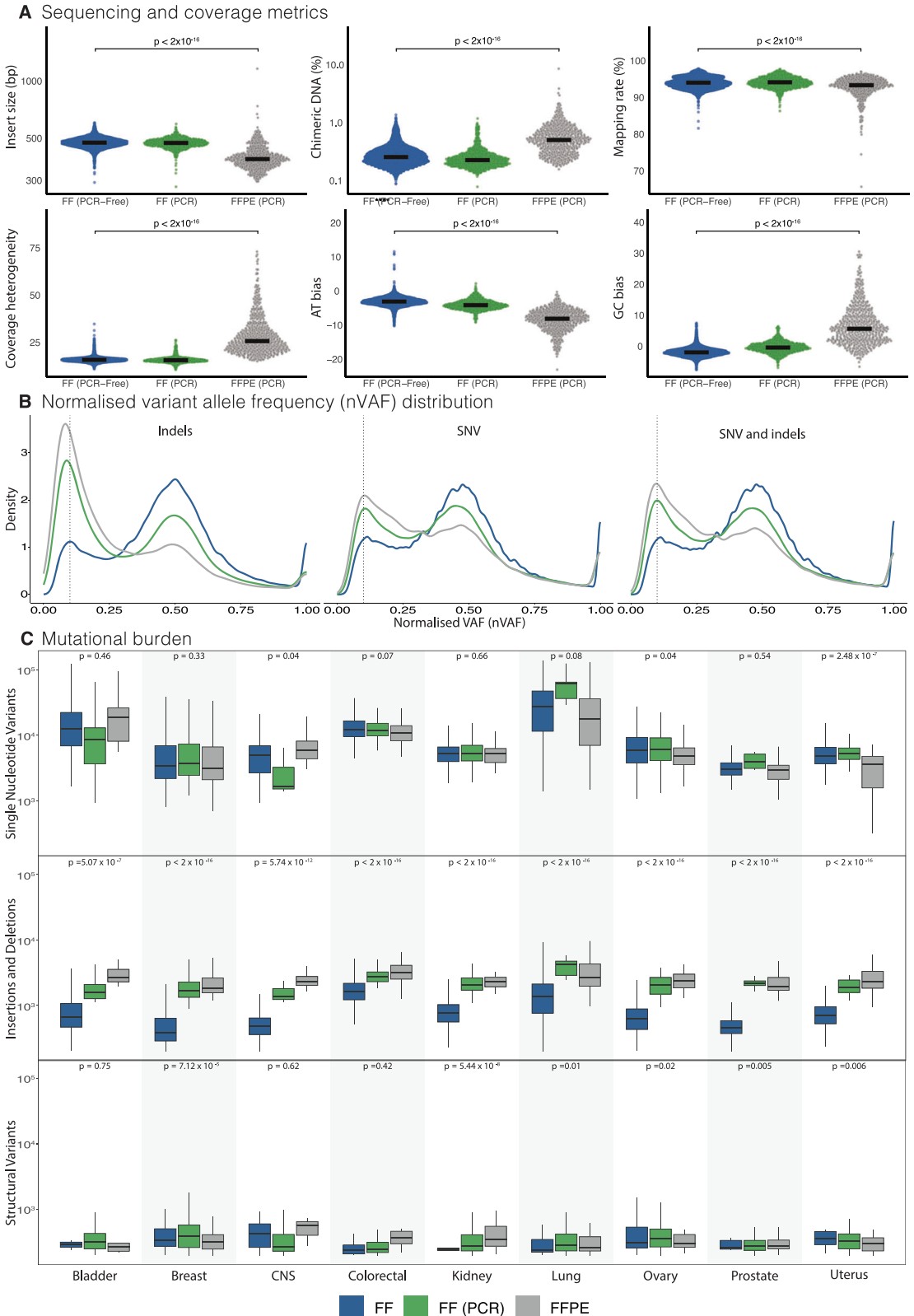

**A** Sequencing and coverage metrics

**B** Normalised variant allele frequency (nVAF) distribution

**C** Mutational burden

FF FF (PCR) FFPE

cancers (FF: 6.4%, FFPE: 4.5%), and *CCND1* (FF: 6.9%, FFPE: 8.5%) were comparably retrieved. The calling of driver homozygous deletions was compromised in ~45% of FFPE samples and 34% of FF samples due to an excess of copy number segmentation (or 'hypersegmentation'), a known issue with copy number calling in general, and was not specific to FFPE samples. In the remaining samples, homozygous deletions in three clinically important tumour suppressor genes, *CDKN2A*, *MAP2K4*,

and *PTEN* occurred at analogous percentages in FF and FFPE samples (Supplementary Tables 7 and 8). No actionable gene fusions were identified in this FFPE cohort.

**Mutational signature analysis defines FFPE-related signatures**

Given the extensive size of the FFPE cohort, we were able to detect and comprehensively characterise artefactual mutational signatures. Both

**Fig. 1 | Quality of sequencing data in Genomics England Cohort. A** Comparison of sequencing and coverage metrics between FF ($n = 10,115$), FF(PCR) ($n = 899$), and FFPE(PCR) ($n = 578$). Insert sizes represent lengths of sequenced DNA fragments. Chimeric DNA percentage is the proportion of reads synthesised from more than one template. Mapping rate is the percentage of reads that can be mapped to the reference genome. Coverage heterogeneity is the read depth uniformity across the genome. Adenosine/thymine (AT) and guanine/cytosine (GC) bias indicates the percentage of reads that are under or overrepresented in AT-rich or GC-rich genomic regions. The *p*-values indicate statistical comparisons between FF and FFPE cohorts using a two-sided Wilcoxon rank-sum test. **B** Normalised variant allele frequency distribution. Variant allele fraction (VAF) is the proportion of sequencing reads reporting a specific variant, whilst cancer cell content is the estimated percentage of tumour cells in the sample. The dotted vertical line is located at VAF 0.1

for reference. **C** Comparison of single nucleotide variant, indel, and structural variant mutational burdens across organ types. Bladder (FF $n = 359$, FF PCR $n = 31$, FFPE $n = 10$), Breast (FF $n = 2509$, FF PCR $n = 283$, FFPE $n = 169$), CNS (FF $n = 504$, FF PCR $n = 76$, FFPE $n = 17$), Colorectal (FF $n = 2469$, FF PCR $n = 113$, FFPE $n = 88$), Kidney (FF $n = 1355$, FF PCR $n = 95$, FFPE $n = 30$), Lung (FF $n = 1290$, FF PCR $n = 114$, FFPE $n = 64$), Ovary (FF $n = 527$, FF PCR $n = 60$, FFPE $n = 34$), Prostate (FF $n = 384$, FF PCR $n = 84$, FFPE $n = 98$), Uterus (FF $n = 718$, FF PCR $n = 43$, FFPE $n = 68$). The box and whisker plots in this figure are defined as follows: the centre line represents the median, the bounds of the box indicate the lower (25th percentile) and upper (75th percentile) quartiles and the whiskers extend to the minimum and maximum values. The *p*-values indicate statistical comparisons between FF and FFPE cohorts using a two-sided Wilcoxon rank-sum test. FF fresh frozen, FF (PCR) fresh frozen with polymerase chain reaction, FFPE formalin-fixed paraffin-embedded.

known and previously uncharacterised artefactual patterns were discerned (Supplementary Data 2). Two substitution signatures associated with FFPE were identified (Fig. 3A; Supplementary Data 2). The first, SBS57, was previously postulated to be an artefactual signature[20], although not overtly associated with FFPE. The other, a hitherto undescribed pattern (SBS FFPE), is characterised by T > C mutations (in particular at A$\underline{T}$A, A$\underline{T}$C, C$\underline{T}$T, and T$\underline{T}$T (mutated base underlined)) as well as C > T mutations (in particular at A$\underline{C}$A, A$\underline{C}$T, and T$\underline{C}$T). 64.2% of FFPE samples carried either SBS57 and/or SBS FFPE (371/578). More than half of the FFPE samples contained SBS57 (55.36%, 320). The proportion of SBS57 per sample tended to decrease as the total substitution count increased. By contrast, SBS FFPE was identified in fewer samples (17.12%, 99), appeared to be idiosyncratic, and tended to dominate the mutational landscape of affected samples, presenting with a 'hypermutator' phenotype (Fig. 3B). We were unable to find associations between SBS FFPE and any other feature, such as extraction protocol or tumour-type. SBS57 was also detected in a small fraction of FF samples that underwent PCR during library preparation (Supplementary Table 9 and Supplementary Data 5). The absence of SBS FFPE in these FF samples with PCR suggests that it is a manifestation of FFPE fixation. We additionally found a new pathognomonic indel signature characterised by 1 base pair A/T insertions and deletions at long polyA/T and dinucleotide deletions at long repeats, accompanied by long insertions at non-repeats (Fig. 3C; Supplementary Data 2). This signature, herewith referred to as ID FFPE, was present in 99.7% (576) of FFPE samples and also followed the pattern of decreasing proportional contribution as total indel count increased (Fig. 3D). Notably, all common biological signatures typically found in FF samples were identified in FFPE samples even when artefactual signatures were present (Supplementary Datas 3, 4). There was a systematic misassignment of SBS3 and SBS8, Homologous Recombination (HR)-deficiency-associated substitution signatures, which do not have distinctive peaks and are relatively flat and featureless. These were systematically over-called in samples that do not necessarily have biological HR-deficiency (Fig. 4).

Given the observations above, we created a score to quantify FFPE-related artefact per sample, called FFPEimpact—a quantitative index based on the proportion of FFPE-related substitution and indel artefacts (Supplementary Methods Section S1.8). FFPEimpact scores ranged from 0 to 0.74 with a median of 0.27 (IQR = 0.14) in the cohort of 578 cases. There did not appear to be systematic loss of genomic information as FFPEimpact scores increased and there was no correlation to sequencing metrics. This was consistent across all tissue-types (Fig. 4). However, FFPEimpact appeared to be enriched in the Covaris-based DNA extraction protocol when compared to a Qiagen system ($p < 0.0001$) (Supplementary Fig. 5). At lower levels, FFPEimpact scores are predominantly influenced by ID FFPE, with its contribution increasing as the FFPEimpact score increases. However, at higher scores, there is a discernible contribution from both indel and SBS artefacts (Supplementary Fig. 4). Crucially, FFPEimpact can be

provided as a metric within a typical WGS report to enhance clinical awareness that a particular FFPE WGS sample may contain substantial artefacts, informing clinical interpretation.

The identification of artefactual signatures within any sample enables the subtraction of these signatures prior to the application of clinically relevant algorithms. For example, HRDetect[21], a probabilistic score of Homologous Recombination deficiency, can be critically salvaged providing biologically valuable information (Fig. 4; Supplementary Methods Section S1.9; Supplementary Data 6). Note that 18 samples were identified as HRDetect 'high' or HR-deficient, in contrast to 331 samples that were misassigned with SBS3 and SBS8.

The results thus far indicate that it is possible to analyse WGS of FFPE samples using customary 'primary' (alignment) and 'secondary' (somatic mutation-calling) WGS bioinformatic pipelines without filtering of variants based on VAF or any other parameter, minimising the risk of discarding clinically relevant variants. Instead, existing WGS bioinformatic pipelines can be used with the simple characterisation of FFPE signatures and the addition of the FFPEimpact score. Subsequent subtraction of FFPE-related signatures during downstream 'tertiary' analysis can be seamlessly incorporated if desired, and algorithmic stratification based on mutational signatures such as HRDetect can be utilised effectively.

### Analysis of validation cohorts
To validate this potential informatic process, 51 matched pairs of FF and FFPE samples were sourced from the Oxford Molecular Diagnostics Centre[14]. This cohort included patients with breast, colorectal, kidney, lung, prostate, or uterine cancers (Supplementary Table 2). We applied the same analytic workflow of a standard WGS bioinformatic pipeline followed by FFPEimpact and FFPE-related artefactual subtraction in tertiary analysis. Using our method above, there was no significant difference in the total number of calls in driver genes (Fig. 5A; Supplementary Table 11 and Supplementary Data 10). There was strong concordance in actionable variants between the two preparations ($r = 0.84$, $p < 0.0001$; Fig. 5B; Supplementary Table 12), reiterating that FFPE-derived WGS data can confer clinical value. Similarly strong correlations were seen in substitution signatures ($r = 0.98$, $p < 0.0001$; Fig. 5C) and rearrangement signatures ($r = 0.86$, $p < 0.0001$; Fig. 5C). In keeping with the observed larger indel burden in FFPE samples (Supplementary Table 10; Supplementary Figs. 6, 7), the concordance of indel signatures was only moderate ($r = 0.59$, $p = 0.0081$; Fig. 5C). Copy number solutions for ploidy/aberrant cell fractions were not automatically achieved in ~7.8% (4/51) of FF samples and 19.6% (10/51) of FFPE samples due to low tumour cellularity (Supplementary Data 7). Satisfactory copy number solutions for these samples were subsequently achieved with manual reseeding. 57% (29/51) of the FF/FFPE pairs showed strong concordance across ploidy, aberrant cell fractions and overall copy number profiles. Interestingly, 10% of pairs showed heterogeneity in copy number profiles, believed to be genuine biological differences in copy number aberrations.

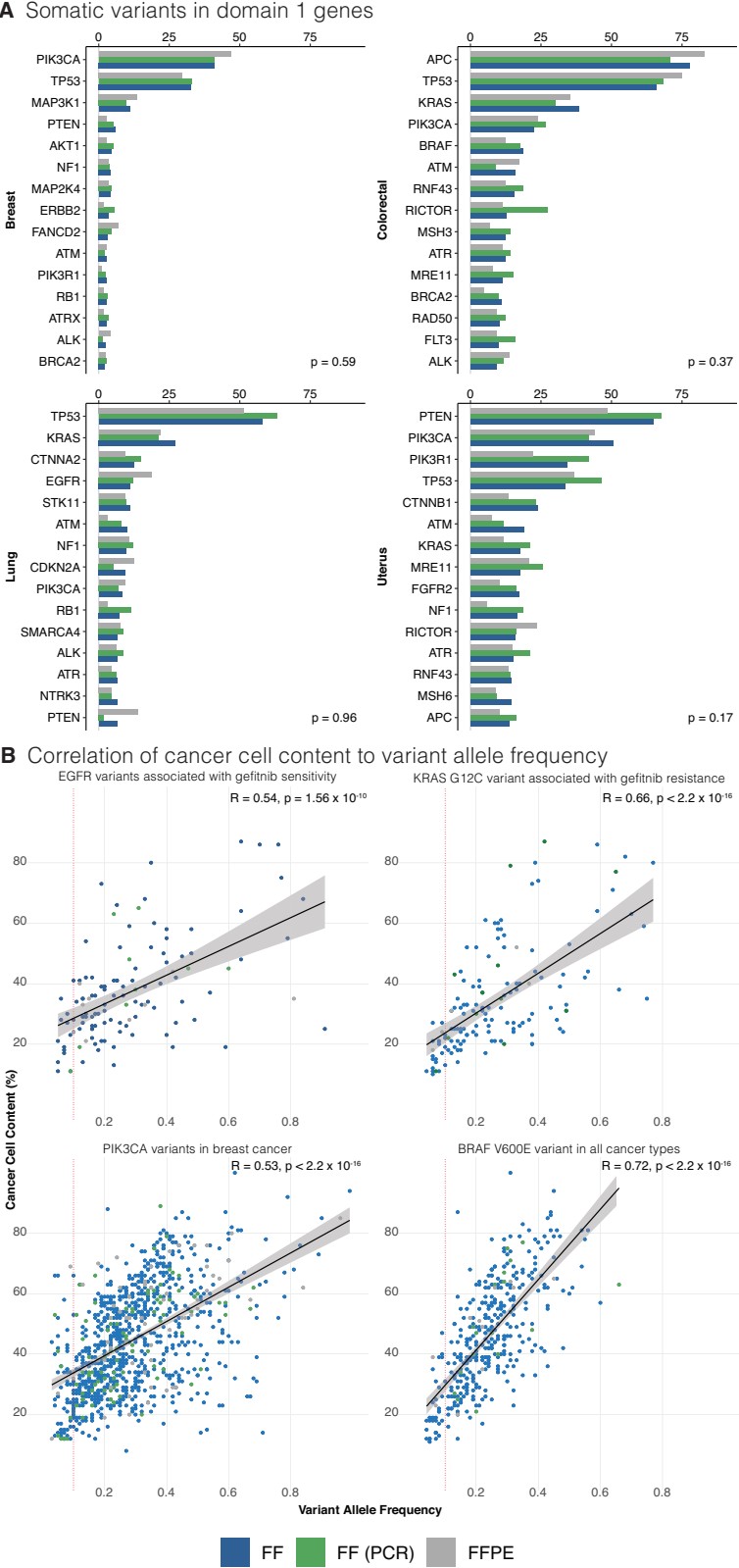

**Fig. 2 | Comparison of putative driver events in Genomics England Cohort.**
**A** Comparison of detection of Domain 1 variants across different sample prepara-
tions (reported as a percentage of samples). Additional organ types are presented
in the supplementary information. Kruskal–Wallis rank sum test was used for sta-
tistical analysis. **B** Comparison of percentage cancer cell content to variant allele
frequency for selected actionable mutations. The solid black line represents the
linear regression fit, and the shaded area around the line indicates the 95% con-
fidence interval of the fit. A vertical red dotted line is demonstrated at variant allele
frequency 0.1 to demonstrate that a significant number of mutations would be

discarded if conventional bioinformatic filtering was applied. Top left panel plots
*EGFR* variants associated with gefitinib sensitivity in lung cancer. Top right panel
plots the *KRAS* G12C variant associated with gefitinib resistance in lung cancer.
Bottom right panel plots *PIK3CA* variants in breast cancer and the bottom left panel
plots *BRAF* V600E variant in all cancer groups included in the study. Correlation
was assessed using Spearman's Rank Correlation (two-sided test). FF fresh frozen,
FF (PCR) fresh frozen with polymerase chain reaction, FFPE formalin fixed paraffin
embedded.

**A** Substitution signatures enriched in FFPE samples

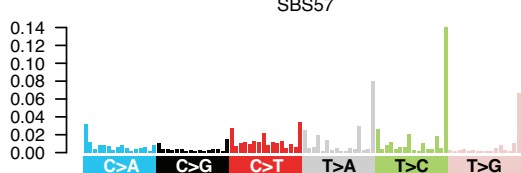 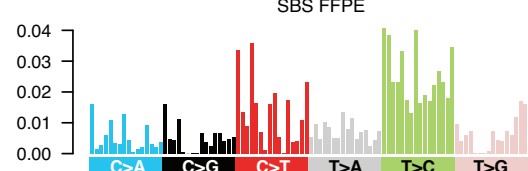

**B** Exposure of substitution signatures

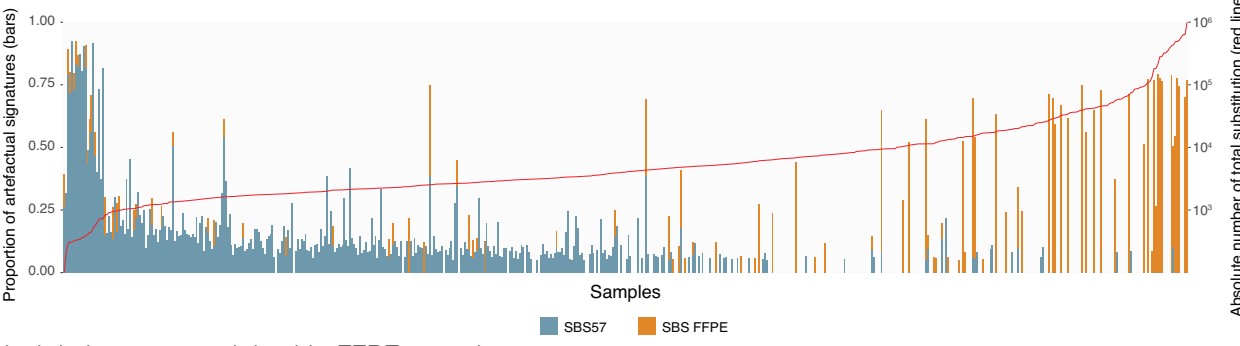

**C** Indel signature enriched in FFPE samples

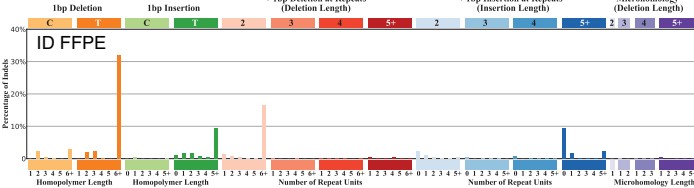
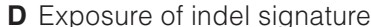

**D** Exposure of indel signature

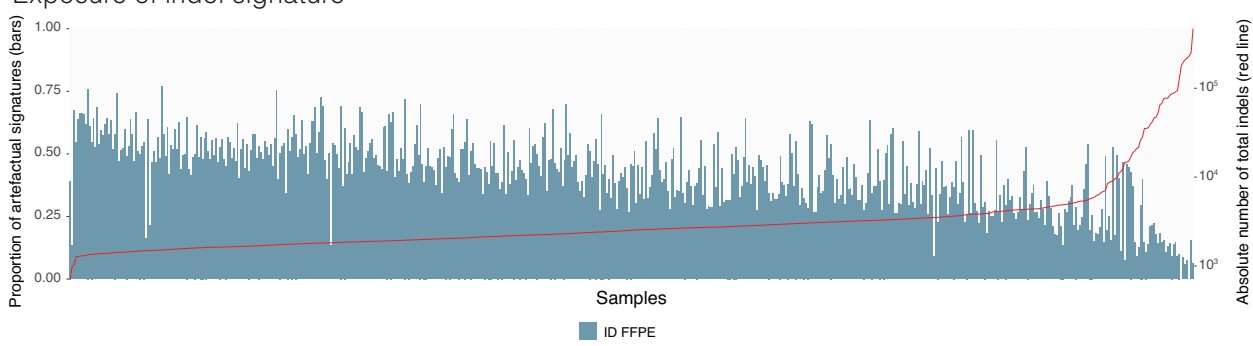

**Fig. 3 | Mutational signatures associated with FFPE artefact. A** Substitution signature profiles for SBS57 and SBS FFPE using a 96-channel format. **B** Exposure of artefactual substitution signatures in Genomics England Cohort. The bar chart represents exposure as a proportion of total substitutions per sample, with samples arranged in order of total substitution burden and the red line represents the total number of substitutions in each sample. **C** Indel signature profile for ID FFPE using the COSMIC indel 83-channel format. **D** Exposure of artefactual indel signature in Genomics England Cohort. The bar chart represents exposure as a proportion of total indels per sample, with samples arranged in order of total substitution burden and the red line represents the total number of indels in each sample. FFPE formalin-fixed paraffin-embedded.

Indeed, 13.7% (7/51) FF samples performed slightly worse than FFPE samples, correlating to differing tumour cellularity between FF and FFPE samples. These findings are in-keeping with previous reports of heterogeneity in results between samples taken from different sites within the same tumour[22–24]. Importantly, quantification of artefact using the FFPEimpact score was again possible (Supplementary Data 8) underscoring the dominant role of indel artefact, yet highlighting the increasing significance of SBS artefact with higher scores.

A second validation set was sought from a real-world prospective neoadjuvant clinical trial of early-stage triple-negative breast cancer patients (PARTNER), which provided 14 samples with FFPE WGS data and matched FF WGS data (co-consented to the Personalised Breast Cancer Programme (PBCP) in Cambridge (herewith, PARTNER/PBCP)

(Supplementary Table 3). Sample processing and analysis were akin to what would occur in routine clinical diagnostic pathways. The majority of identified genomic characteristics were concurrently identified in both specimen preparations. Note that while some variants were seen in FF samples and not in FFPE, the reverse is also true (Fig. 5D; Supplementary Table 13 and Supplementary Data 10). Using customary mutational signature assignment methods, there was good concordance of positively and negatively identified substitution (~91%), indel (~88%), and rearrangement signatures (~89%). Where there were discordances in signature assignments, the absolute assignment was of a very small number of mutations (reflected in the pale colouring in Fig. 5D). Indeed, for indel signatures, there was a higher likelihood of systematic false positive assignment of ID6 (related to TOP2A

## Overview of 578 formalin-fixed paraffin embedded cancer specimens

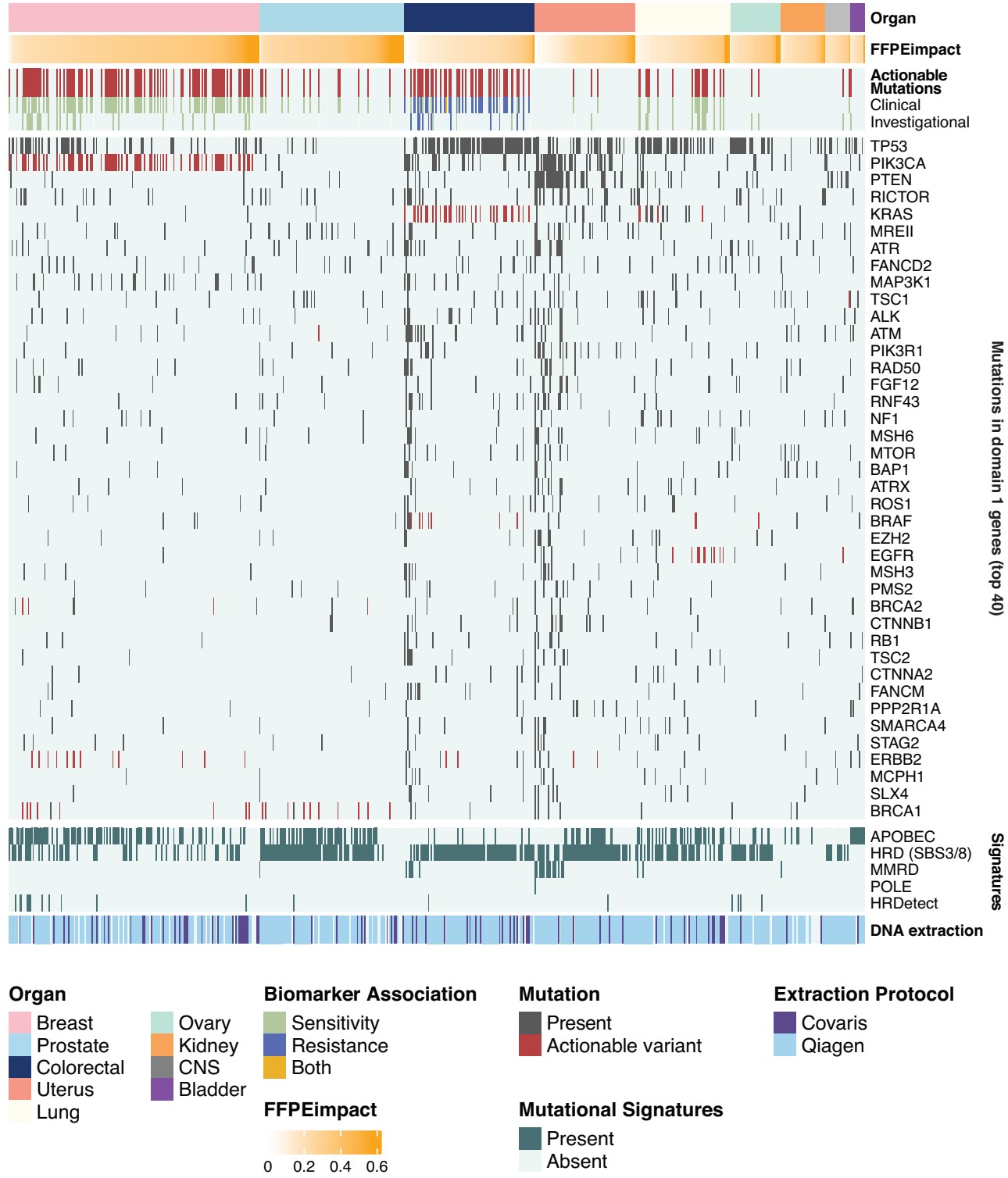

**Fig. 4 | Overview of Genomics England FFPE cohort.** Samples are arranged by organ type and then by FFPEimpact score (in ascending order). Mutations were considered actionable if they met the criteria discussed in the methodology section. Actionable mutations were divided into Clinical and Investigational depending on ESCAT tier (see the "Methods" section). Mutations in the top 40 mutated Domain 1 genes are presented with actionable variants highlighted. Mutational signatures were grouped by aetiology. APOBEC: APOBEC protein family dysfunction; HRD: homologous recombination deficiency (SBS3 and SBS8); MMRD mismatch repair deficiency, POLE DNA polymerase epsilon dysfunction, HRDetect HRDetect algorithm corrected for artefact as discussed in methodology and a supplemental appendix. DNA extraction protocol compares the two different protocols (Covaris and Qiagen) used for the Genomics England cohort. FFPE formalin-fixed paraffin-embedded.

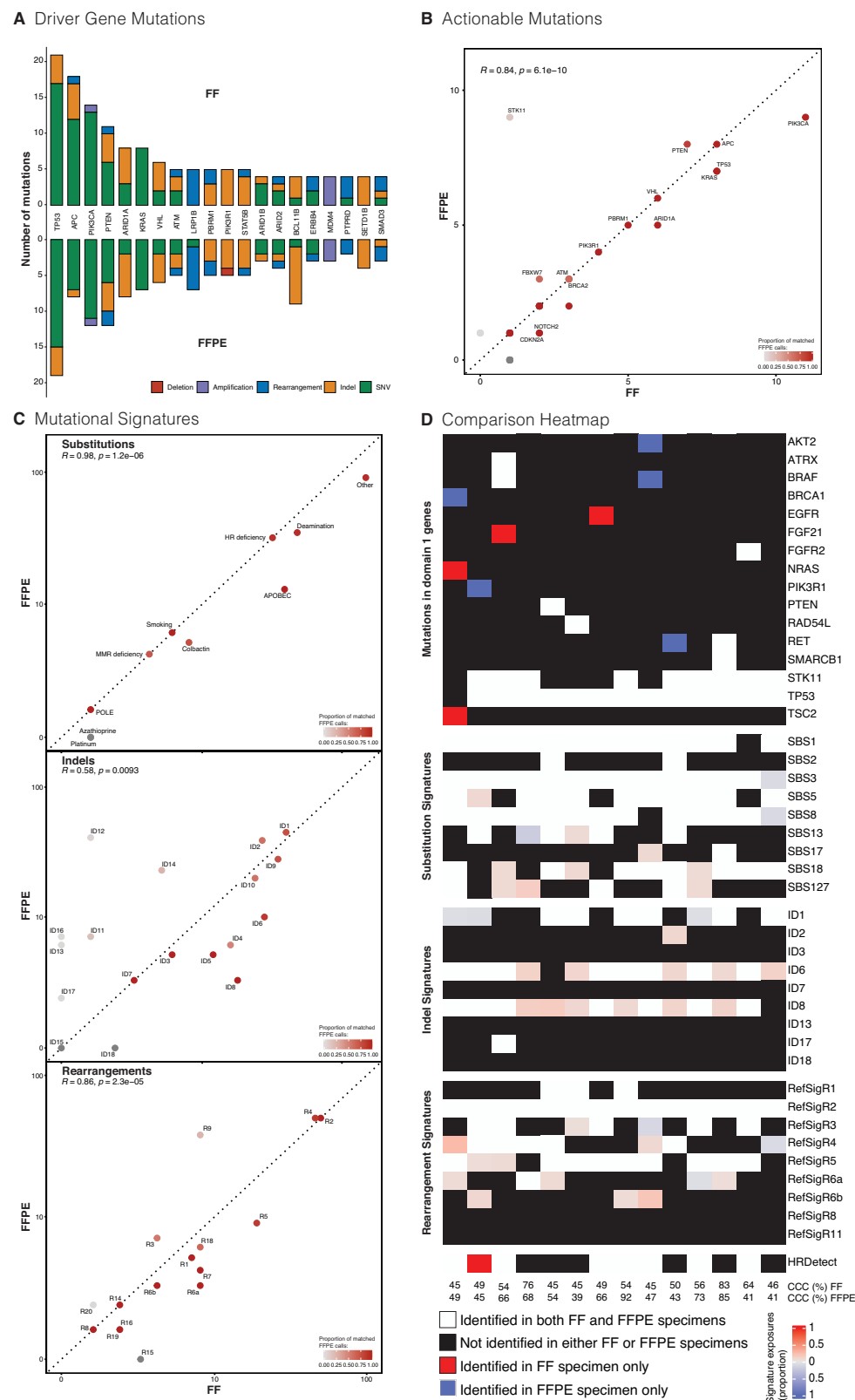

**A** Driver Gene Mutations

**B** Actionable Mutations

**C** Mutational Signatures

**D** Comparison Heatmap

mutations) and ID8 (related to homologous-recombination deficiency) in FF samples. However, it is preferable to use multiple genomic features rather than individual signatures to classify cancers, as exemplified by the HRDetect algorithm. We found that all of the samples that had ID8 reported in FF samples but not in FFPE samples did not have a high HRDetect score, reinforcing that the ID signature assignments that were unique to FF were likely false positives.

Critically, quantification of artefact per sample followed by subtraction of artefact as described above helped to improve WGS analysis, demonstrating 7/8 concordance of the HRDetect homologous-recombination score as an example (Supplementary Data 9). Interestingly, comparing HRDetect with other HRD callers like CHORD[25] underscores the value of using FFPEimpact to remove ID artefact signatures (Supplementary Data 8). Unlike HRDetect, CHORD was

**Fig. 5 | Comparison of two validation cohorts with matched FFPE and FF specimens. A** Comparison of count and class of driver gene mutations in the Oxford Cohort ($n = 51$) (supporting data Table S11). Concordance in calling of **B** actionable variants (supporting data Table S12) and **C** mutational signatures between FFPE and FF specimen in the Oxford Cohort. Dotted line represents 100% concordance ($R = 1$), and the colour of the points demonstrates the proportion of FFPE calls that are complementary to the matched FF specimen. A log-scale is used for mutational signatures. **D** Heatmap comparing genomic characteristics in the PARTNER/PBCP

Cohort ($n = 14$). The top 20 mutated Domain 1 genes are presented. Common breast signatures are presented for substitution and rearrangement, whilst only indel signatures with proposed aetiology are presented. HRDetect is again calculated following correction for indel artefact as discussed in the "Methods" section and supplementary information. Cancer cell content (CCC) is provided for both FF and FFPE samples. Correlation was assessed using Spearman's Rank Correlation. FF fresh frozen, FFPE formalin fixed paraffin embedded.

misled by FFPE-induced indels, resulting in no HR deficiency calls (Supplementary Data 9).

## Discussion

Although FF-derived material undoubtedly offers WGS data of the highest quality, the logistical demands of FF collections remain a significant operational barrier for many, exacerbating disparities in health-care access. Recent commissions highlighting such inequalities have advocated for breakthroughs that can reduce the access gap and democratise diagnostics[26–28].

Whilst FF tissue may be the gold standard for WGS, FFPE remains the preferred method of tissue preservation as histology and immunohistochemistry are central to diagnostics. Moreover, tissue yields from certain clinical diagnostic procedures, such as core biopsies, may be too small to obtain both FF and FFPE samples. Excisional biopsies of incorrectly presumed benign lesions and/or failure of conventional treatment could lead to ex post facto justifications for WGS, yet further acquisition of FF samples may not be possible. Thus, a sweeping exclusion of FFPE specimens from WGS pathways would systematically exclude patients from access to WGS technology that could confer clinical benefit. Other tissue storage solutions, such as RNAlater[18] may address some of the operational barriers seen with FF tissue and produce higher-quality WGS data than FFPE. However, as long as mainstream tissue diagnostics is reliant on FFPE specimens, there will be a cohort of patients for which only FFPE tissue is available.

In medicine, we regularly encounter situations (or specimens) that are imperfect, and we must do the best we can with the information at hand. Here, we show that although WGS from FFPE samples may sometimes demonstrate poorer quality sequencing metrics, the data obtained are not void of clinical value. Rather than adopting the conventional practice of simply filtering mutations based on their VAF, we sought to derive a means of quantifying and characterising the artefacts, that can then be incorporated into downstream processing to maximise information recovered from FFPE samples[16,17]. We comprehensively characterised artefacts associated with FFPE samples and demonstrated that it is possible to preserve actionable information from WGS of cancers using FFPE specimens, including actionable drivers, mutational signatures, and machine-learning-based algorithms. We developed and validated a new feature—the FFPEimpact score—serving as *caveat emptor* for the interpreting clinician. It is anticipated that with future optimisation of library kits for FFPE WGS, new artefact signatures may be discovered, which can be incorporated into the FFPEimpact score through modifications. Our matched FF and FFPE sample analyses demonstrate that in spite of FFPE-related substitution and indel artefacts and/or deficiencies in copy number signals, there is sufficient concordance in actionable information to support how clinically beneficial WGS data can be obtained from FFPE specimens, if all other options for obtaining tumour material were unavailable (an overview comparison between FF and FFPE WGS is provided in Supplementary Table 14).

The feasibility of WGS in FFPE solid tumours shown here is predicated on having matched normal samples derived from blood, not from surrounding FFPE tissue. The latter incurs a high failure rate. We appreciate that sample collection bias may contribute to the observed high success rate, however, in the validation clinical trial, patients were recruited based entirely on clinical inclusion criteria without sample

preselection. Here, we still found a high success rate and a strong correlation between FFPE and FF WGS samples. It is also important to note that reliable detection of structural variants requires sufficient insert sizes, which may be compromised in older FFPE samples. In this work, we do not address the impact of FFPE on other 'omic' modalities, such as the transcriptome, methylome or proteome, which are increasingly being used in standard-of-care cancer profiling for selected cancer types.

Nevertheless, the findings presented here demonstrate that despite artefacts, WGS can be performed on FFPE specimens if necessary. The analytical advancements presented here can be applied to existing WGS cancer pipelines to characterise the presence of FFPE-associated artefacts and to provide a measure of the amount of artefact using the FFPEImpact score within a WGS report. Crucially, permitting FFPE WGS will not require changes to existing clinical pathways, nor should it interfere with existing FF or RNAlater sample collection. It does, however, widen participation and improve access, and should be in the arsenal of diagnostic tools available to clinicians worldwide.

## Methods

### Study design and participants

Evaluation of FFPE WGS was performed on 578 samples which were compared to 11,014 FF samples in the discovery phase. All patients were diagnosed with cancer in England's National Health Service and underwent a primary surgical procedure with curative intent without prior adjuvant therapy. All provided written informed consent for WGS of tumour and a matched germline sample (peripheral blood mononuclear cells) via the 100,000 Genomes Project and were recruited across all thirteen Genomic Medicine Centres in England. The following tumour-types were included in the study: breast, central nervous system (CNS), colorectal, kidney, lung, ovary, prostate, uterus, and bladder (Supplementary Table 1; Supplementary Fig. 2). A fraction of FF tumour samples underwent polymerase chain reaction (PCR) during library preparation ($n = 899$, 8% of FF samples, herewith referred to as FF(PCR)).

In the validation phase, 51 matched pairs of FF and FFPE samples were sourced from the Oxford Molecular Diagnostics Centre[14]. This cohort included patients with breast, colorectal, kidney, lung, prostate, or uterine cancers (Supplementary Table 2). Another validation set was sought from a prospective neoadjuvant clinical trial of early-stage triple-negative breast cancer patients (PARTNER)[29], which provided 14 samples with FFPE WGS data. These same 14 patients had co-consented to the Personalised Breast Cancer Programme (PBCP), which provided FF WGS data. Both these studies were conducted in Cambridge (Supplementary Table 3).

### Procedures

Details regarding nucleic acid extraction, DNA library preparation and genome sequencing are provided in the appendix (Supplementary Methods Section S1). Of particular note, the FFPE samples underwent a two-step DNA repair process (Supplementary Fig. 8): DNA repair step 1 contained Uracil DNA gylocyslase and Endonuclease IV to remove deaminated cytosine residues and hydrolyse the backbone at the abasic site. DNA repair step 2 contained RecJ to remove single-stranded DNA. All sequencing was performed at an accredited

laboratory on a HiSeqX for the Genomics England cohort, HiSeq2500 for the Oxford cohort and a Novaseq 6000 for the PARTNER/PBCP cohort (Illumina, San Diego, CA, USA) to an average coverage of >90× for tumour samples and >30× for matched normal samples (Supplementary Table 4).

Raw short reads were aligned to reference genome build 38. Sequencing quality metrics were analysed and compared between different sample preparations. Insert sizes represent lengths of sequenced DNA fragments (Supplementary Datas 1, 7). Chimeric DNA percentage is a metric that indicates the proportion of reads synthesised from more than one template (caused by random inter-chromosomal DNA cross-linking due to DNA strand breakage). Mapping rate is defined as the percentage of reads that can be mapped to the reference genome. Coverage heterogeneity reports the read depth uniformity across the genome, unevenness is calculated as median for the root square deviation of coverage calculated in non-overlapping 100 kb windows; this metric would be 0 for a genome with absolutely uniform coverage. Adenosine/thymine (AT) and guanine/cytosine (GC) bias indicates the percentage of reads that are under or overrepresented in AT-rich or GC-rich genomic regions. Variant allele fraction (VAF) is the proportion of sequencing reads reporting a specific variant, and cancer cell content is the estimated percentage of tumour cells in the sample.

Somatic variants were identified in the 100,000 Genomes Project cohort according to the pipeline detailed by Genomics England, version 8 (Supplementary Methods Section S1.5), without additional VAF-based filtering. For validation cohorts, FF and FFPE pairs were contrasted using the same bioinformatic pipeline: somatic single nucleotide variant (SNV) detection was performed with CaVEMan (Cancer Variants through Expectation Maximisation: http://cancerit.github.io/CaVEMan/) and indel detection used Pindel (http://cancerit.github.io/cgpPindel/)[30], again without additional VAF-based filtering. This permits artefacts to be detected as "variants", but through the size of the cohort, we were able to characterise these artefacts more effectively.

Somatic variants are defined by Genomics England as variants in a virtual panel of potentially actionable genes (168 genes listed in Actionable genes in solid tumour v2; Supplementary Methods Section S1.5). They are akin to 'driver' events that are causally implicated in tumourigenesis, although it cannot be excluded that a subset of these mutations may be passenger events. The fraction of samples reporting Domain 1 SNVs was calculated per organ for each DNA preservation method and compared. The relative abundance of SNVs in Domain 1 genes was assessed for each organ type. The top fifteen Domain 1 genes with the most variants in the FF samples were identified for each organ and the proportions of samples reporting these variants were compared between the different specimen preparation methods. The correlation between VAF and cancer cell content was assessed for specific variants to understand whether there were systematic differences between cohorts and potentially false-positive calls in FFPE samples. Somatic copy number drivers of two classes, amplification of an oncogene and homozygous deletion of a tumour suppressor gene, were sought in a set of eight clinically important genes (*CDKN2A, MAP2K4, PTEN, ERBB2, FGFR1, GNAS, SOX2, CCND1* - full methodology detailed in the Supplementary Methods Section S1.10.).

Mutations were considered actionable if identified as ESMO Scale of Clinical Actionability for molecular Targets (ESCAT) tier I or tier II in the OncoKB (https://www.oncokb.org, v.3.12) or Clinical Interpretation of Variants in Cancer (CiVIC, https://civicdb.org/, v.2.0.0) databases. Detailed definitions of tier I and tier II variants are available elsewhere[31]. Briefly, ESCAT tier I targets are ready for implementation in routine clinical decisions; alteration-drug match is associated with improved outcomes in clinical trials. ESCAT tier II are investigational targets; the alteration-drug match is associated with antitumour activity, but the magnitude of the benefit is unknown.

Mutational catalogues were generated for all FFPE samples as described previously[20]. Briefly, single base substitution (SBS) variants

were classified according to their trinucleotide context, forming a 96-channel catalogue for each sample. Similarly, indel (ID) catalogues were generated for each FFPE (PCR) sample using 83 channels according to indel type and length. Tissue-specific SBS signatures or existing ID signatures were assigned for all FFPE (PCR) samples as previously reported[32]. In short, tissue-specific common SBS signatures were identified for each sample and subtracted from the mutational catalogue. For ID catalogues, existing ID signatures were identified for each sample and subtracted. Residual mutations were then clustered to identify recurrent patterns for signature extraction related to FFPE artefact. For each organ, the set of putative SBS signatures was annotated using the reference signatures reported at https://signal.mutationalsignatures.com/. The identified artefactual signatures were combined to calculate their contribution towards the total mutational catalogue per sample.

## Statistical analysis
Statistical differences between groups were calculated in R (version 4.0.3) using Wilcoxon rank-sum and Kruskal–Wallis statistical tests. Correlation was assessed using Spearman's Rank Correlation. All numerical data are reported as median (interquartile range).

## Reporting summary
Further information on research design is available in the Nature Portfolio Reporting Summary linked to this article.

## Data availability
Primary data from the 100,000 Genomes Project, which are held in a secure research environment, are available to registered members of the Research Network. Membership of the Research Network is open to all individuals, students, or staff affiliated with UK academic research institutions, NHS trusts, relevant charitable organisations, foreign universities and research institutions, governmental departments, and foreign healthcare organisations involved in significant research activity. Review of applications can take up to 10 working days. Following approval, confirmation of affiliation, and passing of information governance training, access to the Research Environment can take up to 2 working days. See https://www.genomicsengland.co.uk for further information or contact M.A.B., Chief Scientific Officer at Genomics England (matt.brown@genomicsengland.co.uk). Source data are provided with this paper.

## Code availability
Analysis was performed in R (version 4.0.3). Mutational signature analysis was performed using the signature.tools.lib package (https://github.com/Nik-Zainal-Group/signature.tools.lib.git). FFPEimpact code for artefact has been made available online (https://github.com/Nik-Zainal-Group/FFPE_impact, https://doi.org/10.5281/zenodo.12725299).

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

## Acknowledgements

This work was funded by Cancer Research UK (CRUK) Advanced Clinician Scientist Award (C60100/A23916, S.N.-Z.), Dr. Josef Steiner Cancer Research Award 2019 (S.N.-Z.), Basser Gray Prime Award 2020 (S.N.-Z.), CRUK Pioneer Award (C60100/A23433, S.N.-Z.), CRUK Grand Challenge Award (C60100/A25274, S.N-Z.), CRUK Early Detection Project Award (C60100/A27815, S.N.-Z.), and the National Institute of Health Research (NIHR) Research Professorship (NIHR301627, S.N.-Z.). This work was also supported by the NIHR Cambridge Biomedical Research Centre (BRC-1215-20014 and NIHR203312). The views expressed are those of the author(s) and not necessarily those of the NIHR or the Department of Health and Social Care. The Personalised Breast Cancer Programme (PBCP) received funding from the following bodies: Addenbrookes Charitable Trust Grant [9800]; The Mark Foundation for Cancer Research and Cancer Research UK [C9685/A25177]; Cancer Research UK [A27657]. The PARTNER trial received funding from the following: Cancer Research UK [CRUKE/14/048]; AstraZeneca [1994-A093777]. This work was enabled by the access to data and findings generated from the 100,000 Genomes Project under the auspices of the Pan-Cancer GeCIP (project RR239). The 100,000 Genomes Project is managed by Genomics England Limited (a wholly owned company of the Department of Health and Social Care), funded by the National Institute for Health Research and NHS England. The Wellcome Trust, Cancer Research UK and the Medical Research Council have also funded research infrastructure. The 100,000 Genomes Project uses data provided by patients and collected by the National Health Service as part of their care and support. The PBCP and PARTNER trials acknowledge primarily the patients and the families and friends who supported them in participating in these studies. Also to NIHR Cambridge BRC for their support for staff and infrastructure costs; The Cancer Research UK Cambridge Centre Cambridge for their support for staff and infrastructure costs; Cancer Molecular Diagnostics Laboratory for their support for sample collection; The Precision Breast Cancer Institute Team for their support for sample collection; The Cambridge Clinical Trials Centre—Cancer Theme for their support with core staff support; The clinical trials support staff at all participating sites.

## Author contributions

Study conception, design, or data acquisition: S.N.-Z., S.Bas., L.H., G.E., P.R., C.C., J.A., A.S., L.J., M.T., M.B., L.G. Data curation, analysis and/or interpretation: S.Bas., L.H., A.D., G.C.C.K., S.Bel., Y.M., Z.K., H.R.D., G.R., L.C., D.B. Visualisation: S.Bas, L.H. Manuscript drafting: S.Bas, L.H., S.N.-Z. All authors were involved in manuscript review and editing. Manuscript critical revision: S.Bas, S.N.-Z.

## Competing interests

A.D., H.R.D., G.C.C.K., G.R., and S.N.-Z. hold patents or have submitted applications on clinical algorithms of mutational signatures: MMRDetect (PCT/EP2022/057387), HRDetect (PCT/EP2017/060294), clinical use of signatures (PCT/EP2017/060289), rearrangement signature methods (PCT/EP2017/060279), clinical predictor (PCT/EP2017/060298), and hotspots for chromosomal rearrangements (PCT/EP2017/060298). Z.K.

is an employee of Illumina, Inc. The remaining authors declare no competing interests.

## Additional information

## PARTNER Trial Group

Jean Abraham ⓘ [3,4]

## Personalised Breast Cancer Program Group

Jean Abraham ⓘ [3,4], Steven Bell ⓘ [3,4] & Carlos Caldas[4]

A full list of members and their affiliations appears in the Supplementary Information.

