## [Peer Review File · Nature Communications]

Large-Scale Analysis of Whole Genome Sequencing Data from Formalin-Fixed Paraffin-Embedded Cancer Specimens Demonstrates Preservation of Clinical UtilityThis manuscript has been previously reviewed at another journal that is not operating a transparent peer review scheme. This document only contains reviewer comments and rebuttal letters for versions considered at *Nature Communications*.

REVIEWER COMMENTS

Reviewer #1 (Remarks to the Author):

The authors have made an important effort to clarify most of the points raised in the previous review. However, there are some aspects that are still confusing. The main value of the manuscript may be the support to the use of routine pathology material in formalin fixed paraffin embedded (FFPE) for genomic studies. However, the possible limitations and ways to deal with them in the clinical practice and how the information generated in this study can be used to improve the analysis of sequences from FFPE are not completely clear. I think there are two issues particularly relevant, one is how to use the information of the new FFPE artifact sequencing signatures to improve the interpretation of the sequences from these tissues. Related to this point, is there any particular approach that can be derived from this study to improve the analysis of the sequences from this material? Second, the concordance between the results in fresh frozen (FF) and FFPE samples in the validation studies of the two cohorts in which paired samples of both types of tissues were available need some clarification. I think that more information on these issues would enhance the practical value of the information.

1) The finding of new signatures related to FFPE treatment of the samples is of interest but it is not clear how this may help in the clinical practice to improve the analysis of genomic sequences. Apparently, the FFPEimpact score gives an idea of the quality of the sequences from this material, but is there any way to use the information in the analysis to select the variants? is it a score that would lead to disregard some cases with low quality sequences? Are these signatures only valuable in the analysis of mutational signatures of the genomes?

2) In response to our point 1.2 on the concordance between FF and FFPE mutational data, the authors indicate in the rebuttal letter that "The global concordance in the PARTNER/PBCP 14-cohort is very high (96%). Of the domain 1 genes presented, 9/224 (4%) are discordant". I am not sure where these numbers come from. In figure 5D and Table S13 the number of mutated genes is 38 and the concordance between FF and FFPE tissues is 76%. However, if mutations of TP53 and SGK11 are excluded the concordance is only 43% (7 out of 16). The issue of mutations in SGK11 is particularly intriguing because in the validation "Oxford cohort" the concordance between FF and FFPE was only 10% (Figure 5B and Table S12). However, in the Partner cohort the apparent mutational frequency of SGK11 is particularly high (64% 9/14), higher than previously reported.

3) The relationship between the data in Figure 5A and Table S12 "Matched Actionable Mutations in Oxford Cohort" is not clear. I may miss something but the number of alterations of several genes in the figure do not match with those in the table. For instance, TP53 SNV in the figure are 17 and 15 in FF and FFPE in the figure whereas 8 and 7, respectively, in the table. Similar apparent discordances in other genes. Are not the same alterations represented in figure 5A and 5B?

4) I would suggest that in addition to clarify the issues indicated in the two previous points, the authors should provide individual mutations in supplementary tables, at least of the validation cohorts. It may be important to see if particular alterations in the genes are the same in both types of material. In addition, it is not clear why the authors did not follow the recommendation of confirming by an orthogonal approach the alterations identified in some samples in the FF and FFPE material as suggested in our point 1.2. The authors indicate that samples from the Oxford cohort are not available, what is certainly a limitation. However, they say they do not have the ethic consent from the Partner cohort to perform targeted analyses. It is difficult to understand they had the consent to perform large scale genomic sequencing and not to have the consent to validate the results.

5) The authors claim that there were no differences between key clinical mutations between FF and FFPE. But for some genes with actionable mutations the authors report a higher incidence of EGFR gefitinib sensitive mutations in FFPE (14.1%) compared with 8% in FF samples. It may not be statistically significant but worth saying a word of caution. Could be this difference due to a different proportion of adenocarcinoma and squamous cell carcinomas in FF and FFPE samples? Did the authors consider different histological subtypes of tumors? There is an apparent discordance

between the text and Figure 2A regarding EGFR mutations. In the text the authors say that gefitinib sensitivity variants were more frequent in FFPE samples but in the figure it seems to show higher frequency of variants in the FF samples. Does this mean that there were differences in other EGFR mutations?

6) The authors need to double check some of the mutations they indicate in the text, particularly the one referring to EGFR and gefitinib sensitivity. In line 159 the authors state "gefitinib sensitivity (L353R, G719S or codon9 deletions)". Do the authors mean p.Leu858Arg (p.L858R) in exon 21 of EGFR instead of L353R? codon19 deletions, should not be exon 19 deletions? Please double check others.

7) The authors discuss that in the validation series the worst correlation of CNV between FF and FFPE may be due to tumor heterogeneity, but this was not the case for SNV mutations in the same cohort. Do the authors mean that the tumor heterogeneity was affecting more CNVs than SNV?

8) The pages in the supplementary appendix are not numbered. It is difficult to find the information referenced in the text. For instance, I do not find the list of the 168 genes included as domain 1 that the authors indicated they are in page 7 of the appendix, or the comparison inter groups indicated to be in appendix page 10 (answer to point 1.1). Other references to the appendix are also difficult to follow.

Reviewer #2 (Remarks to the Author):

I am satisfied with the authors' addresses and comments.

Their new Table in Point 2.12 should be included in this paper, because it is oriented by cancer clinics.

Point-by-point response to reviewer:

Again, we thank the reviewers for their time reviewing our manuscript. We have done our utmost to address specific comments and provided clarity, which we hope will facilitate the peer review process. Below, we provide a point-by-point response (blue text) to Reviewer comments (black text). Text that will be inserted into the manuscript are marked in red text.

Reviewer #1:

Remarks to the Author

The authors have made an important effort to clarify most of the points raised in the previous review. However, there are some aspects that are still confusing. The main value of the manuscript may be the support to the use of routine pathology material in formalin fixed paraffin embedded (FFPE) for genomic studies.

However, the possible limitations and ways to deal with them in the clinical practice and how the information generated in this study can be used to improve the analysis of sequences from FFPE are not completely clear.

I think there are two issues particularly relevant, one is how to use the information of the new FFPE artifact sequencing signatures to improve the interpretation of the sequences from these tissues. Related to this point, is there any particular approach that can be derived from this study to improve the analysis of the sequences from this material?

Second, the concordance between the results in fresh frozen (FF) and FFPE samples in the validation studies of the two cohorts in which paired samples of both types of tissues were available need some clarification. I think that more information on these issues would enhance the practical value of the information.

Thank you for taking the time to review our revised manuscript. We appreciate the thoughtful feedback and are pleased to see that most of the previous concerns have been addressed. We welcome this opportunity to further clarify our manuscript.

We agree that the primary goal of our study is to support cancer whole genome analysis from FFPE-derived material, crucially, in situations where no other tissue samples are available. This context is important because, at present, there are no clinical pathways that permit whole genome sequencing (WGS) using FFPE specimens. In real-world clinical situations, we are increasingly encountering situations where FFPE is the only available material, and many patients are denied access to an investigation that could lead to improved outcomes. Our key message is that there is sufficient preservation of clinical utility in FFPE-derived WGS data, to reassure clinicians and funders to enable wider access to WGS.

With regards to analysis of sequencing data from FFPE, our manuscript has introduced a conceptual shift compared to the conventional approach used in the community. Previous convention for analysing FFPE WGS data included filtering mutations with a VAF <10% - this would result in exclusion of true variants, thus there was concern regarding the clinical utility of WGS FFPE data for many years. Our approach does not depend on any pre-filtering at all - This is new. Because we took this approach, it allowed all us to “see” the FFPE artefact signatures. We believe that mutational signatures have the potential to be leveraged clinically. Specifically, we have created an FFPE-impact score that can help to inform a clinical user about the general state of an FFPE sample, when they are interpreting the WGS report in a multi-disciplinary team meeting.

We hope that we have provided the appropriate clarification and now improved the sign-posting to the right tables and appendices for the concordance/validation analyses.

Response to other specific points below:

1) The finding of new signatures related to FFPE treatment of the samples is of interest but it is not clear how this may help in the clinical practice to improve the analysis of genomic sequences. Apparently, the FFPEimpact score gives an idea of the quality of the sequences from this material, but is there any way to use the information in the analysis to select the variants? is it a score that would lead to disregard some cases with low quality sequences? Are these signatures only valuable in the analysis of mutational signatures of the genomes?

Thank you for highlighting that the new signatures related to FFPE treatment is of interest.

In the third results section “Mutational signature analysis defines new FFPE-related signatures”

- The first paragraph describes the new signatures
- The second paragraph describes how we created the FFPE impact score based on the new signatures, and how we assessed it.
- And at the end of the second paragraph (line 247-250) we say “FFPEimpact can be provided as a metric within a typical WGS report to enhance clinical awareness that a particular FFPE WGS sample may contain substantial artefacts”. It is simply a quantitative indicator that the sample is not in its an ideal state, serving as a caveat in clinical interpretation of WGS.

No, the FFPE-related mutational signatures and thus the FFPEimpact score are a valuable indicator of the generic state of the sample. Note that this is not unusual in clinical settings. When pathological specimens are received and need to be reported, it is sometimes the case that the material is of substandard quality. A histopathological report is still provided, but the report will state the caveat that the quality of the material is not of the highest level. What we are providing with the FFPEimpact score is the same principle, with the added value that it is a quantitative value for the level of damage. We have also added in the final paragraph of the discussion: “The analytical advancements presented here can be applied to existing WGS cancer pipelines, to characterise the presence of FFPE-associated artefacts and to provide a measure of amount of artefact using the FFPEImpact score within a WGS report.”

Our manuscript does not advocate or suggest that a certain cut-off for FFPEimpact is used to disregard specific samples as global clinical utility was preserved with all scores. However, it is not uncommon that following clinical adoption, guidelines such as this may be introduced in the future when analysing individual samples separately.

2) In response to our point 1.2 on the concordance between FF and FFPE mutational data, the authors indicate in the rebuttal letter that “The global concordance in the PARTNER/PBCP 14-cohort is very high (96%). Of the domain 1 genes presented, 9/224 (4%) are discordant”. I am not sure where these numbers come from. In figure 5D and Table S13 the number of mutated genes is 38 and the concordant between FF and FFPE tissues is 76%. However, if mutations of TP53 and SGK11 are excluded the concordance is only 43% (7 out of 16). The issue of mutations in SGK11 is particularly intriguing because in the validation “Oxford cohort” the concordance between FF and FFPE was only 10% (Figure 5B and Table S12). However, in the Partner cohort the apparent mutational frequency of SGK11 is particularly high (64% 9/14), higher than previously reported.

How we arrived at 4%

When considering the clinical utility of FFPE generated WGS data, we must address the concerns that data generated may introduce both false positives and false negatives. Thus, our approach to considering concordance took both of these into account. In Table S13/Figure 5D we analysed 16 genes and there were 14 samples (16 x 14 = 224). The concordance looked at whether a mutation was:

- Called by both FF and FFPE (concordant)
- Called by neither FF nor FFPE (concordant)
- Called by FF only (non-concordant)

- Called by FFPE only (non-concordant)

There were 215 concordant calls vs 9 non-concordant calls hence our 4% figure.

TP53

We are unclear what the issue is:

- If the point is that TP53 is over-represented in PARTNER/PBCP-14: That is because the 51 samples from the Oxford cohort were from a range of tumour types (breast, colorectal, kidney, lung, prostate and uterus) while the PBCP/PARTNER cohort (n=14) were all triple negative breast cancers (TNBC). TP53 mutations are common somatic drivers in TNBC (Nik-Zainal et al. 2016, Staaf et al. 2019) and thus its omission from analysis would obviously alter the concordance as we would likely be removing an important clonal driver for this cohort.
- That said, it is not clear to us why removing TP53 would be a sensible thing to do in this analysis, *regardless of tumor-type*. It is a driver event. We want to know what the concordance is between FF and FFPE for important drivers. Exclusion of TP53 does not make sense.

STK11

The actionable mutation analysis for the Oxford cohort (Figure 5B and table S12) looked at all classes of mutation (substitutions, indels, copy number variants and rearrangements). We found that:

- 1/51 paired samples had an STK11 mutation in both FF and FFPE (concordant)
- 42/51 paired samples did not have STK11 mutations in neither FF nor FFPE (concordant)
- 8/51 paired samples had STK11 mutations in FFPE but not in FF (non-concordant)

Thus, similar to previous definition of concordance, STK11 status in the Oxford cohort was 84% concordant. Addressing the discordant STK11 mutations in FFPE samples, all were copy number - related deletions) with no specific tumour type predilection. It is probable that this is due to copy number issues associated with FFPE as alluded to in the manuscript :

Line 291:

“Copy number solutions for ploidy/aberrant cell fractions were not automatically achieved in ~7.8% (4/51) of FF samples and 19.6% (10/51) of FFPE samples due to low tumour cellularity. Satisfactory copy number solutions for these samples were subsequently achieved with manual reseeded. 57% (29/51) of the FF/FFPE pairs showed strong concordance across ploidy, aberrant cell fractions and overall copy number profiles..”

With regards to the PARTNER/PBCP cohort – we agree that the STK11 mutation frequency is higher than previously reported but first they are all of one tissue type (unlike the Oxford cohort) and second our sample size is also much smaller (n = 14). Thus any attempted explanation of these types of differences should be with caution. The main point of this section of the manuscript is within cohort concordance between FF and FFPE. It is not an assessment of inter-cohort differences.

3) The relationship between the data in Figure 5A and Table S12 “Matched Actionable Mutations in Oxford Cohort” is not clear. I may miss something but the number of alterations of several genes in the figure do not match with those in the table. For instance, TP53 SNV in the figure are 17 and 15 in FF and FFPE in the figure whereas 8 and 7, respectively, in the table. Similar apparent discordances in other genes. Are not the same alterations represented in figure 5A and 5B?

Perhaps the reviewer has mixed up the tables and figures.

- **Figure 5A** reporting ‘driver mutations’ corresponds with data in **Table S11**.
- **Figure 5B** reporting ‘actionable mutations’ (which are a subset of driver mutations because not all driver mutations are clinically actionable), corresponds with data in **Table S12**.

To ensure that there will be no further misunderstanding, we have referenced the corresponding table in the figure legend for each Figure panel.

Definition of actionable mutations is noted in line 494 of the “procedures” section.

4) I would suggest that in addition to clarify the issues indicated in the two previous points, the authors should provide individual mutations in supplementary tables, at least of the validation cohorts. It may be important to see if particular alterations in the genes are the same in both types of material. In addition, it is not clear why the authors did not follow the recommendation of confirming by an orthogonal approach the alterations identified in some samples in the FF and FFPE material as suggested in our point 1.2. The authors indicate that samples from the Oxford cohort are not available, what is certainly a limitation. However, they say they do not have the ethic consent from the Partner cohort to perform targeted analyses. It is difficult to understand they had the consent to perform large scale genomic sequencing and not to have the consent to validate the results.

We have added 2 tables to the supplementary tables (LT24 and LT25) that provide the individual variants for the validation cohorts that can be referenced by our readers.

With regards to the request to perform orthogonal confirmation by performing targeted analysis of samples presented in this manuscript – this work has been achieved through collaboration across multiple teams. Some of the cohorts are clinical translational recruitments (e.g. PBCP) and others are clinical trials (e.g., Partner). In other words, these are not research samples that are stored in our freezers with blanket ethical approval to tap into the samples as desired. Each clinical study has its own ethical approval and protocol. It is not in our gift to simply do extra targeted panel sequencing post-hoc as it is not in the protocol of work:

1. This is by far the largest scale analysis of whole genome sequenced FFPE cancer samples in the literature – achieving this required us to establish research collaborations with multiple groups namely Genomics England, PBCP and Partner groups in Cambridge (distinct projects) and the research group in Oxford. We are grateful to our partners for the collaboration but must respect and abide by the protocols and ethics in place for each respective clinical sampling pathway and protocol.
2. We are *analysts* in this enormous process. Sequencing data has been made available to us in an anonymised format. We do not have the luxury of going back to samples and re-doing some experiments.
3. Whilst a comparison between targeted panels and WGS is interesting, it is not the purpose of this manuscript. The purpose of our paper is to support the use of FFPE tissue for WGS when no other tissue sample is available. Pathways for WGS from FF samples exist and targeted panels are not usually performed as validation of the findings. The preservation of clinical utility and sufficient concordance is enough for the manuscript to achieve its aims.
4. Orthogonal confirmation suggests that a targeted panel would be able to definitively resolve any discrepancy between the FF and FFPE calls but this is an oversimplification. It ignores other biological confounders for the slight variance observed (such as intratumour heterogeneity).

5) The authors claim that there were no differences between key clinical mutations between FF and FFPE. But for some genes with actionable mutations the authors report a higher incidence of EGFR gefitinib sensitive mutations in FFPE (14.1%) compared with 8% in FF samples. It may not be statistically significant but worth saying a word of caution. Could be this difference due to a different proportion of adenocarcinoma and squamous cell carcinomas in FF and FFPE samples? Did the authors consider different histological subtypes of tumors? There is an apparent discordance between the text and Figure 2A regarding EGFR mutations. In the text the authors say that gefitinib sensitivity variants were more frequent in FFPE samples but in the figure it seems to show higher

frequency of variants in the FF samples. Does this mean that there were differences in other EGFR mutations?

Thank you for your comment. Unfortunately, we do not have information about the different histological subtypes but agree that it could be a potential explanation. We have added the following statement to the manuscript:

Line 169:

The marginally higher proportion of these EGFR variants in FFPE samples might be attributed to variations in the histological subtypes of lung cancer. However, this observation should be interpreted with caution due to the limited sample size.

Apologies for the confusion. When changing the colours in Figure 2A (as recommended by another reviewer) the FF and FFPE colours in the bar chart were mistakenly swapped. This has now been corrected and demonstrates that in the lung cancer EGFR mutations are proportionally more frequent in the FFPE cohort – as reported in the manuscript.

A subset of all EGFR mutations are associated with gefitinib sensitivity, not all of them. Hence, our next sentence states:

“As actionable mutations are of immediate clinical value, specific variants were explored.

For example, variants associated with gefitinib response in lung cancers were not differentially represented in FFPE-derived WGS samples. *EGFR* variants indicating gefitinib sensitivity (L858R, G719S, or exon 19 deletions) were present in 8.1% (104) of FF lung samples, 8.6% (9) of FF (PCR) samples and 14.1% (9) of FFPE samples respectively.

The EGFR mutations relevant to gefitinib sensitivity are a subset of the total EGFR variants and are not what is in Figure 2A.

6) The authors need to double check some of the mutations they indicate in the text, particularly the one referring to EGFR and gefitinib sensitivity. In line 159 the authors state “gefitinib sensitivity (L353R, G719S or codon9 deletions)”. Do the authors mean p.Leu858Arg (p.L858R) in exon 21 of EGFR instead of L353R? codon19 deletions, should not be exon 19 deletions? Please double check others.

Thank you. We have made these corrections and proof-read the manuscript to ensure all other variants are accurate.

7) The authors discuss that in the validation series the worst correlation of CNV between FF and FFPE may be due to tumor heterogeneity, but this was not the case for SNV mutations in the same cohort. Do the authors mean that the tumor heterogeneity was affecting more CNVs than SNV?

In the manuscript we discuss that 57% of the FF/FFPE pairs in the Oxford cohort showed strong concordance. We agree that this is a lower figure than we see with SNVs and discuss some of the reasoning (lines 291-301). In general, copy number analysis is particularly challenging for FFPE samples, this is a well-known problem in the community. It is likely due to the level of fragmentation of DNA.

8) The pages in the supplementary appendix are not numbered. It is difficult to find the information referenced in the text. For instance, I do not find the list of the 168 genes included as domain 1 that the authors indicated they are in page 7 of the appendix, or the comparison inter groups indicated to

be in appendix page 10 (answer to point 1.1). Other references to the appendix are also difficult to follow.

Apologies for this oversight. This occurred with change of versions. We have now added page numbering to the supplementary appendix and updated all page references in the manuscript. A link to the list to all Domain 1 genes has been added to the supplementary appendix. The table referred to is Table S6. We have updated all references to the appendix to make it easier for readers to follow.

REVIEWERS' COMMENTS

Reviewer #1 (Remarks to the Author):

The authors have addressed all my questions and concerns. I thank them for their interest.